# Transfer Learning for Segmentation Problems: Choose the Right Encoder and Skip the Decoder

## Abstract

It is common practice to reuse models initially trained on different data to increase downstream task performance. Especially in the computer vision domain, ImageNet-pretrained weights have been successfully used for various tasks. In this work, we investigate the impact of transfer learning for segmentation problems, being pixel-wise classification problems that can be tackled with encoder-decoder architectures. Given a U-Net architecture, we find that transfer learning the decoder does not help downstream segmentation tasks, while transfer learning the encoder is truly beneficial. Overall, the advantageous effect of pretrained models is strongest in low-data regimes. Our investigation is therefore motivated by a real world medical image (binary) segmentation problem, where labeled data is scarce and we study the model performances in such low-data regimes.

We exemplify within our experimentation framework that pretrained weights for a decoder may yield faster convergence, but they do not improve the overall model performance as one can obtain equivalent results with randomly initialized decoders. However, we show that it is more effective to reuse encoder weights trained on a segmentation or reconstruction task than reusing encoder weights trained on classification tasks. Our findings suggest that model pretraining on large-scale segmentation datasets can provide encoder weights that are more suitable for downstream segmentation tasks than an encoder pretrained on the ImageNet classification task. We also propose a contrastive self-supervised approach with multiple self-reconstruction tasks, which provides encoders that are suitable for transfer learning in segmentation problems in the absence of segmentation labels.

## 1 Introduction

Transfer learning and reusing pretrained weights is common practice when training deep learning models. Reusing weights from a model that was pretrained on a large scale dataset often has several advantages: faster training time, reduced costs, ecological footprint and improved performance - especially in low-data regimes. For computer vision problems, the use of ImageNet-pretrained weights for model initialization has de facto become standard practice as these encode information related to the visual content of millions of images from diverse domains. To better leverage the underlying structure in the data, advanced self-supervised learning methods such as SimCLR (Chen et al., 2020a), BYOL (Grill et al., 2020), and MoCo (He et al., 2020) have been proposed in the last years. Common to all approaches is that they only pretrain an encoder network. In order to capture fine-grained visual features in the latent representations, the ConRec framework (Dippel et al., 2021) extends upon SimCLR by incorporating a decoder network and jointly optimizing a contrastive and a self-reconstruction loss.

For segmentation tasks, encoder-decoder architectures often reuse ImageNet weights as an initialization for the encoder, whereas the initial weights of the decoder are commonly initialized by drawing from a random distribution: Minaee et al. (2021) surveyed more than 100 recent image segmentation algorithms, and they state that "many people use a model trained on ImageNet" as the encoder part of the network, and re-train their model from those initial weights. In the context of medical image segmentation, it has been observed that self-supervised pretraining with a reconstruction task has the potential to significantly improve the model performance on downstream segmentation tasks (Zhou et al., 2019; Haghighi et al., 2020).

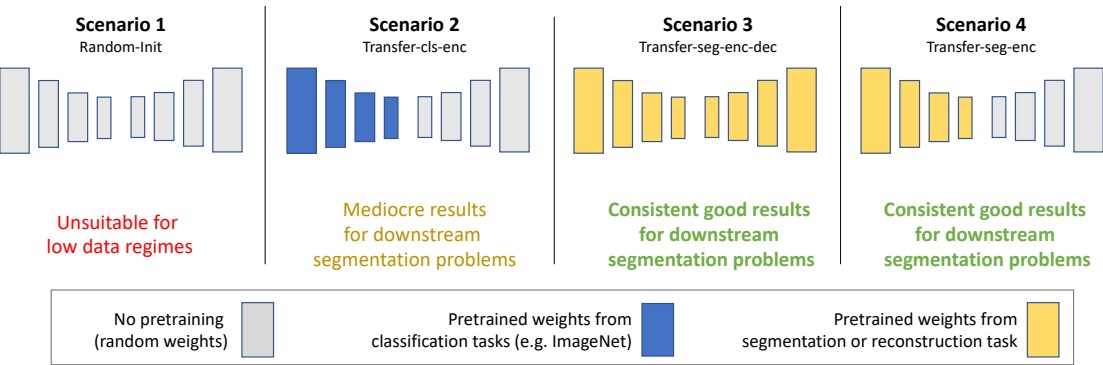

Figure 1: Transfer learning scenarios for segmentation problems. Note that Scenario 2 is commonly applied in literature and our experimental results show that this is suboptimal.

It is yet not well understood, though, which underlying factors are driving the reported improvements.

This study aims at systematically analyzing the effect of different pretraining and initialization approaches for encoder-decoder networks on downstream segmentation tasks. Hereby we limit our experimental setup to encoder-decoder architectures as they are most prominently applied for image segmentation. The competing initialization approaches comprise (a) encoder pretrained on a pretext classification task, (b) entire network pretrained on a pretext reconstruction or segmentation task, and (c) encoder pretrained on a pretext reconstruction/segmentation task with randomly initialized decoder – see Fig. 1. Clearly, these approaches require different levels of annotations, i.e., no labels, class labels, or segmentation masks, which has direct implications on their practical applicability. A comprehensive quantitative analysis of the different learning scenarios is provided for image segmentation tasks of varying sizes derived from four distinct data sources. To simplify and focus our analysis, we limit our experiments to the well-known U-Net architecture (Ronneberger et al., 2015) which is widely used in medical segmentation tasks.

Our experiments confirm the intuition that it is most beneficial to pretrain on pretext tasks that are of the same type as the downstream task. Hence, choosing by default an encoder that was simply pretrained on ImageNet classification as initialization may, in general, be suboptimal for downstream image segmentation problems. Interestingly, we observe that in our experimental setup, the pretraining of the decoder seems to be only of minor importance. Thus, it may be of key importance to carefully select encoder weights that are used for initialization, whereas the decoder branch can be easily learned from scratch on the downstream task. Next to the quantitative analysis of different transfer learning scenarios for segmentation problems, we also provide a qualitative analysis that further explains the observed patterns. We compare encoder representations from different models, and we find that an encoder which is trained with a segmentation pretext task provides fundamentally different representations than an encoder from a classification pretext task.

## 2 Evaluation Environment

Below, we provide a detailed description of the evaluation environment that was used to investigate several transfer learning scenarios.

### 2.1 Datasets

We used six benchmark datasets: Flower Segmentation (Nilsback & Zisserman, 2008), Cityscapes (Cordts et al., 2016), Oxford IIIT Pets (Parkhi et al., 2012) and three subsets of the Pascal VOC dataset (airplanes, cats, horse) (Everingham et al.) – see also Fig. 2 and Table 1 for further details. As those tasks are rather easy, we vary the amount of training data and observe the effect during evaluation. We vary the fraction samples used for training (1%, 5%, 20%) for all datasets and use the remaining samples for evaluation.

Table 1: Dataset size of our six benchmark datasets.

| Dataset | Flowers 17 | Oxford Pets | Cityscape Bus | VOC airplane | VOC cat | VOC horse |
|---------|-----------|-------------|---------------|--------------|---------|-----------|
| #Samples | 849 | 7375 | 483 | 178 | 250 | 147 |

(a) Oxford Flowers 17          (b) Cityscape Bus          (c) Oxford IIIT Pets

(d) Pascal VOC airplane          (e) Pascal VOC cat          (f) Pascal VOC horse

Figure 2: Samples from the six segmentation datasets: Flowers, Cityscape, Oxford IIIT Pets and Pascal VOC airplane, cat, horse with their respective annotations.

## 2.2 Related work – SimCLR, Reconstruction and ConRec

We want to compare popular self-supervised learning methods such as the SimCLR framework (Chen et al., 2020a;b), MoCo (He et al., 2020) and BYOL (Richemond et al., 2020) that only train an encoder with methods that also train a decoder. SimCLR learns representations by learning that views from the same image are similar and that views from different images are dissimilar. Reconstruction-based methods commonly train a model to predict the content of artificially masked parts of an image by optimizing a $l_2$ reconstruction loss (Pathak et al., 2016; Zhou et al., 2019; Dippel et al., 2021). To overcome information redundancy in images, recent approaches suggest masking a very high portion of random patches in order to create a self-supervised task for learning expressive visual features (He et al., 2021; Bao et al., 2021; Zhou et al., 2021b; Bachmann et al., 2022). The ConRec framework (Dippel et al., 2021) combines the contrastive task of the SimCLR framework with a reconstruction task to initialize the decoder. Classification networks tend to extract features which are invariant within a certain class while reconstruction networks extract features capable of capturing fine-grained visual details of a given instance. In order to balance intra-class invariance and feature richness, the ConRec framework jointly optimizes a contrastive and a reconstruction loss. Several recent studies have been following a similar approach (Li et al., 2020; Zhou et al., 2021a), to tackle "feature suppression" of fine-grained details.

The vanilla ConRec model as described in Dippel et al. (2021) has shown promising results for classification tasks. Segmentation tasks require pixel-level precision and therefore it is necessary to exploit fine-grained visual details from an image.

In order to tailor the ConRec approach towards segmentation problems and to build more robust and generalized representations in the decoder, we extent this concept: instead of a single reconstruction task, we challenge the ConRec model with multiple reconstruction tasks while sharing most of the weights between the different tasks. The four tasks include the normal reconstruction of the masked image, segmenting the locations where masks are applied to the image, reverting the image from the color jittered colorspace to the original color space and highlighting the masked regions. Given one example image, the four target images for each reconstruction task are shown in Fig. 3. Subfigure (a) shows an example input image and (b) - (e) show the respective reconstruction targets. (b) depicts the normal reconstruction of the image, (c) also

includes the projection to the original color space, (d) highlights the regions in the image where a mask can be found, and (e) fills the masks with black color. The first three blocks of the decoder are shared between the reconstruction tasks, and the last decoder block is unique to each task.

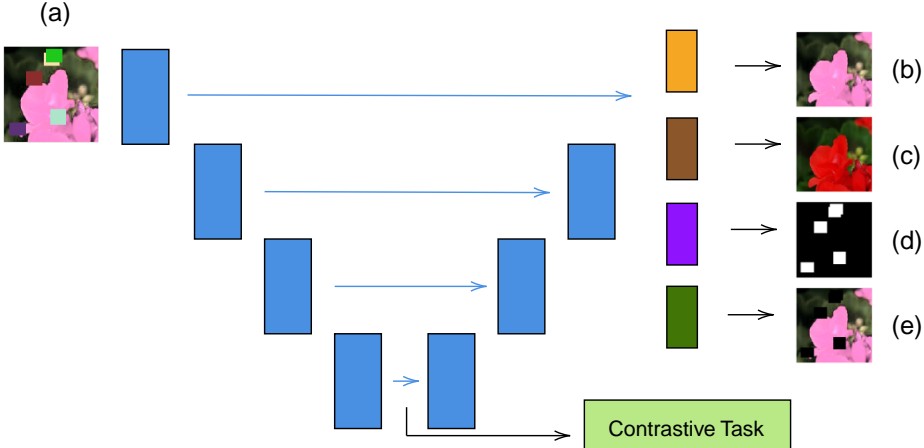

Figure 3: Four reconstruction tasks that are performed simultaneously when training the ConRec framework. Given the input image (a), the ConRec model outputs the images (b)-(e) with its four reconstruction heads.

Table 2: Finetuning results for various pretrained models and scenarios on the Oxford Flowers 17 and Oxford Pets datasets. The model pretraining was performed on the entire unlabeled dataset. To exemplify the impact of the pretraining, smaller subsets of the data (1%, 5%, 20%) were used with binary segmentation labels for finetuning. Displayed values are the dice coefficient on the evaluation set.

| | | Flowers | | | Pets | | |
|---|---|---|---|---|---|---|---|
| Scenario | Model | 1% (8) | 5% (42) | 20% (170) | 1% (74) | 5% (368) | 20% (1475) |
| 1 (random-init) | Random | $79.1 \pm 1.0$ | $89.4 \pm 0.8$ | $94.3 \pm 0.0$ | $71.7 \pm 5.2$ | $84.2 \pm 0.9$ | $88.2 \pm 0.3$ |
| 2 (cls-enc) | SimCLR | $77.7 \pm 0.4$ | $88.8 \pm 0.3$ | $94.1 \pm 0.1$ | $79.2 \pm 1.1$ | $85.1 \pm 0.1$ | $88.2 \pm 0.4$ |
| 3 (seg-enc-dec) | Reconstruction | $84.4 \pm 0.6$ | $91.8 \pm 0.2$ | $95.0 \pm 0.1$ | $80.5 \pm 1.5$ | $86.8 \pm 0.2$ | $89.6 \pm 0.3$ |
| 3 (seg-enc-dec) | ConRec | $85.7 \pm 1.8$ | $92.2 \pm 0.4$ | $95.3 \pm 0.1$ | $83.1 \pm 0.5$ | $87.5 \pm 0.6$ | $90.0 \pm 0.1$ |
| 4 (seg-enc) | Reconstruction | $84.6 \pm 0.2$ | $92.0 \pm 0.1$ | $95.0 \pm 0.0$ | $81.2 \pm 0.1$ | $86.9 \pm 0.5$ | $89.5 \pm 0.1$ |
| 4 (seg-enc) | ConRec | $84.7 \pm 0.6$ | $92.1 \pm 0.1$ | $95.3 \pm 0.1$ | $83.3 \pm 0.6$ | $87.2 \pm 0.4$ | $89.8 \pm 0.2$ |

## 2.3 Implementation Details

As a preprocessing step, we resize the images with padding to the desired target size. During training, we randomly flip the image horizontally and take a random crop containing at least 50% of the image while changing the aspect ratio maximally to $(\frac{3}{4}, \frac{4}{3})$ of the original image ratio. We use the dice loss as the objective function. Following the official ConRec implementation[1], our U-Net has four encoder and decoder blocks consisting of two convolutional layers with batch normalization and max pooling/upsampling in the encoder/decoder. This results in a total of 8.65M parameters for the model. While training for a specific subclass from the Pascal VOC or Cityscapes dataset, we use all the images where there is at least one pixel of the target class visible and discard all other images.

## 3 Do we need a Pretrained Decoder?

We evaluate various model training scenarios in order to assess whether or not a pretrained decoder is advantageous for downstream segmentation tasks. Note that the very same model encoder-decoder architecture

---

[1] https://github.com/bayer-science-for-a-better-life/contrastive-reconstruction

is used for all model training scenarios, we only alter the weight initialization. For pretraining the self-supervised models, we use the Oxford Flowers 102 and Oxford Pets dataset. For finetuning, we train and evaluate models for two segmentation datasets (Oxford Flowers 17; Oxford Pets) with different amount of data (1%, 5% and 20%) used for training. Comprehensive results are listed in Table 2. As expected, transfer learning helps in general, hence scenario 1 with all weights initialized randomly yields the worst results.

Using ConRec as pretext task outperforms all other methods across the different data regimes and datasets, highlighting that the multiple reconstruction tasks lead to more general representations which can be generalized in downstream segmentation tasks. It slightly outperforms the models that were initialized with the reconstruction pretext task. With 20 % training data ($\approx$170 training samples), the differences in performance are marginal, and the dice coefficient is high. This is because the segmentation task is relatively easy as the objects appear prominent in the image and sufficient training data is available to overrule the model initialization.

We aim to investigate if the observed performance increases in the low-data regime are the result of using a pretrained decoder. Therefore, we randomize the decoder of the models that were pretrained with a reconstruction task (i.e. scenario 4) and compare the resulting performance to the fully initialized model (i.e. scenario 3). The results show that randomizing the decoder does not have a significant influence on the final performance. This finding was further validated on a real-world dataset – see Section 4.

There are two potential explanations for this finding: Either the computational graph in a pretrained decoder is irrelevant for the downstream segmentation task, or even very little number of labeled training samples are required to restore the data transformations required. Fig. 4a shows the dice coefficient on the test set for the 5% case over the training time. The ConRec model with the random decoder (i.e. ConRec RD) needs more training steps but it can finally achieve the same accuracy than the initialized decoder (i.e. ConRec). Thus, the pretrained decoder is indeed relevant as it yields optimal model performance with less training steps. However, given a larger number of training steps, the model can reach the same performance level – even for low-data regimes.

This observation shows that the random decoder is not the driver for the performance gap to the SimCLR model initialization, indicating that the ConRec model learns representations in the encoder, which are better suited for segmentation tasks. ConRec's embeddings might contain more encoded information about the position of different features that are unnecessary for the contrastive task.

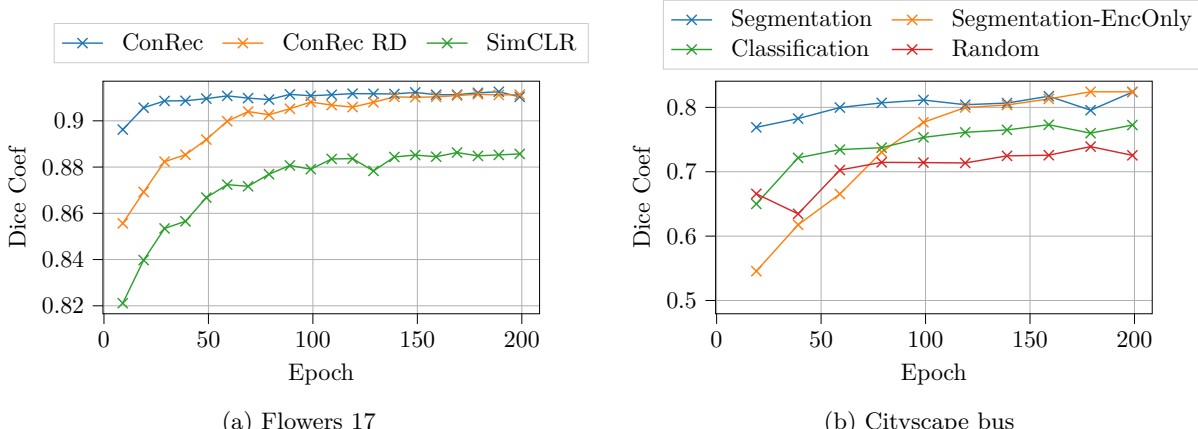

(a) Flowers 17  (b) Cityscape bus

Figure 4: Validation dice coefficient over the training time for two segmentation downstream tasks with different model initializations.

## 4    Implication for Real World Scenario

The results from the previous section indicate that pretraining the decoder has no significant effect on segmentation transfer. We test this hypothesis on a real world scenario where one model is pretrained on a

reconstruction task and then finetuned on a downstream segmentation task. Therefore, we reran the official Model Genesis implementation [2]. Zhou et al. (2019) pretrain their U-Net architecture with 3D volumes of the LUNA lung nodule dataset and then evaluate it on lung nodule segmentation. We reproduce their results and add an additional run where we initialize the decoder randomly. For the random initialization, we only finetune the model one time and for the two other variants, we finetune the models three times and report mean/std dice and IoU performance on the evaluation set.

The results in Table 3 show that there is no significant performance difference measurable between the fully initialized model and the model with a random decoder. This supports the findings on our benchmark datasets.

Table 3: Results from lung nodule segmentation with the Model Genesis initialization compared to the same model with a random initialization. *pretrained encoder; pretrained decoder* replicates the results reported by Zhou et al. (2019) (see Fig. 7, NCS).

| Initialization | Dice coef | IoU |
|---|---|---|
| random encoder; random decoder | 71.54 | 74.47 |
| pretrained encoder; random decoder | $75.20 \pm 1.13$ | $76.94 \pm 0.50$ |
| pretrained encoder; pretrained decoder | $75.10 \pm 0.48$ | $76.78 \pm 0.45$ |

## 5 How to pretrain the encoder for downstream segmentation tasks?

Before, we could show that there is no need to transfer learn a pretrained decoder. However, comparing the results for scenario 2 to the results for scenario 4, we find that the encoder trained with SimCLR performs significantly worse than the encoder trained with ConRec or the Reconstruction task. The SimCLR model does not even bring any advantage over a random initialization.

### 5.1 Segmentation and Classification Pretraining

We want to assess if the limited effect of the pretrained decoder is unique to the reconstruction pretext task or if this also is an observation that occurs when transferring from one segmentation task to another segmentation task. Therefore, we present a transfer learning segmentation experiment in the following.

As the decoder does not have the primary effect on segmentation transfer learning performance, we want to investigate if it is enough to train a classification model and use that for segmentation task transfer. Obtaining classification annotation is often much easier and cheaper than obtaining fine-grained segmentation annotations. Therefore, the insight that pretraining with a classification model also results in good segmentation performance would save much efforts for real-world problems.

We pretrain our U-Net model on one segmentation task and then finetune the model on a different segmentation task. As a pretraining task, we use the Cityscapes dataset (Cordts et al., 2016) with the challenge to segment all cars that appear in the image. We maximize the dice coefficient as the objective function. For pretraining, we extract patches with cars from the dataset with respective ground-truth annotations.

To compare our segmentation pretrained model to a classification model with a comparable training task, we generate datasets of patches showing a scene of the Cityscapes dataset and containing either a car or no car in the images. For every image of the train and validation split of the Cityscapes dataset, we decide by a coin flip if we want to generate a patch with a car or no car. Then, we take a random crop until either no pixels contain a car label (no car) or at least 20% of the pixels have car labels (car). For model selection, we do a random 70/30 train/validation split of the patches. Samples of this dataset are provided in Fig. 5. The model has to predict whether there is a car visible in the patch or not.[3] In both pretraining setups,

---

[2]https://github.com/MrGiovanni/ModelsGenesis
[3]After pretraining the model, we used GradCam activations to validate if the model looks for cars in the image to make classification predictions – see Fig. 5c.

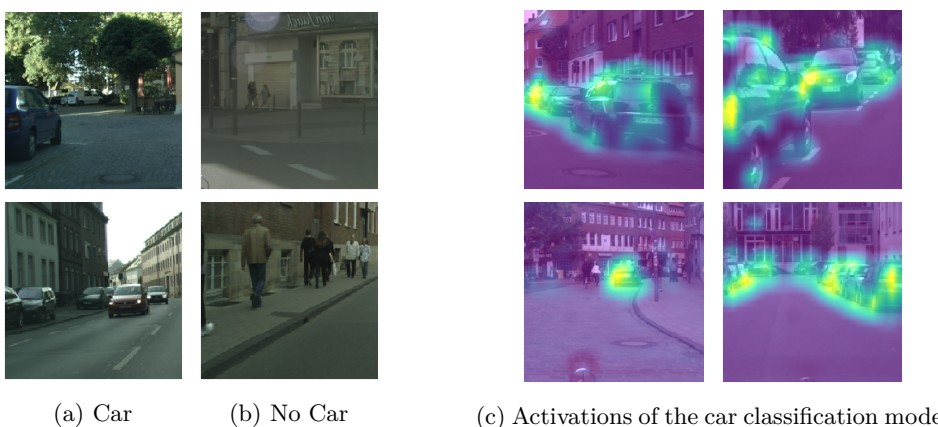

(a) Car          (b) No Car          (c) Activations of the car classification model

Figure 5: Car classification dataset generated by cropping images from the Cityscapes dataset where either a car is present or no car is present on the image.

we used a constant learning rate and optimized the pretraining parameters (learning rate, training epochs, weight decay) towards the best performance on the validation set. We finetune the car segmentation model with an initialized decoder (Segmentation) and a random decoder (Segmentation-EncOnly). Furthermore, we compare these models to the model trained on the car classification task (Classification) and a random initialization (Random). Table 4 also includes the results of different self-supervised approaches that have been pretrained on the whole Cityscape dataset, a U-Net model with an ImageNet initialized Resnet18 encoder and multi-class segmentation pretraining on the SceneParse150 dataset (Zhou et al., 2017).

Table 4: Effect of the decoder when finetuning self-supervised models in comparison to finetuning a classification and a segmentation pretrained model. Displayed values indicate the segmentation accuracy on the evaluation set [dice coefficient].

| Scenario | Model | pretext labels | Bus | Airplane |
|---|---|---|---|---|
| 1 (random-init) | Random | | $75.0 \pm 0.3$ | $65.8 \pm 0.5$ |
| 2 (cls-enc) | SimCLR | | $74.5 \pm 0.6$ | $68.3 \pm 0.9$ |
| 2 (cls-enc) | Classification | cars/noncars | $77.6 \pm 0.9$ | $68.3 \pm 1.2$ |
| 3 (seg-enc-dec) | Reconstruction | | $76.6 \pm 0.6$ | $68.7 \pm 0.8$ |
| 3 (seg-enc-dec) | Segmentation | cars/non-car | $82.3 \pm 0.5$ | $71.0 \pm 0.7$ |
| 3 (seg-enc-dec) | ConRec | | $76.9 \pm 1.0$ | $70.6 \pm 1.7$ |
| 4 (seg-enc) | Reconstruction | | $76.4 \pm 0.2$ | $68.2 \pm 0.2$ |
| 4 (seg-enc) | Segmentation | cars/non-car | $82.0 \pm 0.6$ | $70.2 \pm 0.2$ |
| 4 (seg-enc) | ConRec | | $77.3 \pm 1.1$ | $70.9 \pm 0.8$ |
| 3 (seg-enc-dec) | Segmentation | SceneParse150 | $83.6 \pm 0.2$ | $73.2 \pm 0.3$ |
| 4 (seg-enc) | Segmentation | SceneParse150 | $83.6 \pm 0.1$ | $73.0 \pm 0.3$ |
| 2 (cls-enc) | Classification | ImageNet | $83.5 \pm 0.7$ | $71.2 \pm 0.2$ |

For evaluation, we consider two segmentation problems: bus segmentation and airplane segmentation, as introduced above. Table 4 shows the results after finetuning for both datasets and transfer learning scenarios. The performance gap between the segmentation-pretrained model and random initialization is larger for the bus segmentation task since the bus segmentation task is similar to the pretraining task (i.e. car classification). In analogy to our previous experiments, we cannot observe significant performance differences between the fully initialized model and the model with a random decoder on both datasets. We visualized the dice coefficients over the training time in Fig. 4b. Similar to the observations for self-supervised models, the random decoder model needs more training time but then catches up to the performance of the fully initialized model. The classification model also yields a performance improvement compared to the random initialization but does not reach the same performance as the segmentation model. This observation indicates that although the segmentation decoder does not add much value during transfer learning, the representations in

the encoder transfer better to other segmentation tasks. By comparing the results to the self-supervised model performances on the bus segmentation dataset, we can conclude that the reconstruction based models perform similar to the car classification model but are also significantly outperformed by the model with segmentation pretraining. On the airplane dataset, the ConRec model outperforms the classification and other self-supervised models and performs on par with the segmentation model. This indicates that segmentation and classification pretraining has larger benefits for in-domain pretraining, whereas self-supervised models transfer better to different domains.

The ImageNet initialization of the encoder and the model pretrained on the SceneParse150 dataset outperform all of the other models. This can be explained by the larger amount of pretraining data. However, note that the model pretrained on SceneParse150 is on par on the Bus dataset and outperforms the ImageNet initialization on the Airplane dataset, although being trained on significantly less pretraining data than the ImageNet encoder (20k images vs. 1.2M images). This further gives motivation for a large-scale segmentation pretraining dataset in the order of magnitude as ImageNet.

We perform an additional experiment on the Oxford IIIT Pets dataset (Parkhi et al., 2012) using only the dog images for pretraining. The segmentation model receives the segmentation masks of all dog images (4978 samples) and the different dog breeds (25 classes) serve as labels in the classification task. This scenario better reflects the usual multi-class classification pretraining scheme. During finetuning, we use cat and horse images from Pascal VOC 2012 (Everingham et al., 2010) and compare four model initialization scenarios[4] as shown in Table 5. Similar to the airplane dataset, we use all images from the training set where a cat/horse appears. This yields 131/68 samples for training and 119/79 samples for evaluation. Finetuning with a Resnet18 initialized ImageNet encoder and an initialization from pretraining on the SceneParse150 (Zhou et al., 2017) benchmark is also shown.

Table 5: Finetuning results on the validation set of two Pascal VOC 2012 (Everingham et al., 2010) subsets with different pretraining scenarios [dice coefficient]. Models were either initialized by a segmentation task (Pets dog masks, SceneParse150) or a classification task (Pets dog breeds, ImageNet).

| Scenario | Model | pretext data (#Samples) | VOC cat | VOC horse |
|---|---|---|---|---|
| 1 (random-init) | Random | | $74.2 \pm 1.4$ | $65.0 \pm 1.3$ |
| 2 (cls-enc) | Classification | Oxford Pets, dog breeds (5k) | $76.0 \pm 0.4$ | $70.1 \pm 0.7$ |
| 3 (seg-enc-dec) | Segmentation | Oxford Pets, dog masks (5k) | $83.2 \pm 0.2$ | $79.0 \pm 1.0$ |
| 4 (seg-enc) | Segmentation | Oxford Pets, dog masks (5k) | $82.3 \pm 0.2$ | $77.7 \pm 0.5$ |
| 3 (seg-enc-dec) | Segmentation | SceneParse150 (20k) | $78.8 \pm 0.8$ | $74.4 \pm 0.3$ |
| 4 (seg-enc) | Segmentation | SceneParse150 (20k) | $79.0 \pm 0.1$ | $74.8 \pm 0.3$ |
| 2 (cls-enc) | Classification | ImageNet (1.2M) | $83.2 \pm 0.2$ | $75.3 \pm 0.4$ |

For both downstream segmentation tasks, a pretrained segmentation model significantly outperforms a randomly initialized model as well as a pretrained classification model. This difference is mainly driven by the pretrained encoder – see Scenario 2 vs. 3 and 4. We observe a small difference between the performance of the fully initialized segmentation model and the model with a random decoder. Nevertheless, we observe the additional performance provided by the pretrained decoder to be small compared to the performance gain through the pretrained encoder. These results provide additional evidence that a pretrained segmentation encoder transfers better to downstream segmentation tasks than an encoder pretrained by a classification task.

The dog segmentation initialization performs on par with the ImageNet initialization on the VOC cat datset and gives better performance on the VOC horse dataset whereas the SceneParse150 initialization is outperformed in both cases. This confirms the intuition that transferring between similar tasks (i.e similar image domains) leads to better performance.

The performance difference between classification and segmentation pretraining can also be seen visually. Figure 6 shows segmentation mask predictions after finetuning on the VOC cat dataset. We observe models

---

[4]random initialization (1); encoder pretrained with a classification (2) or segmentation (4) task; encoder-decoder pretrained with a segmentation task (3)

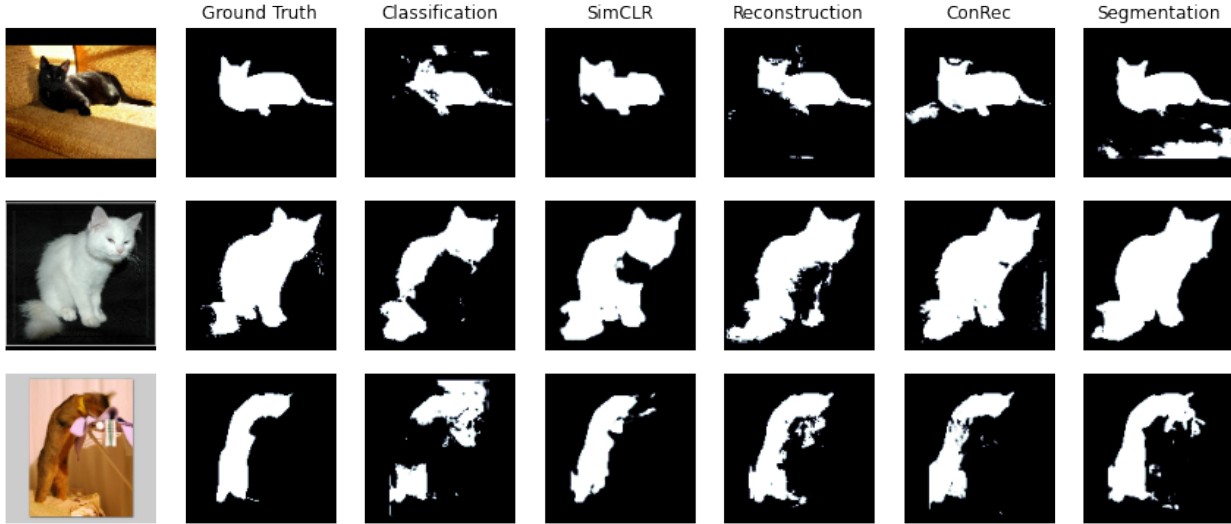

Figure 6: Segmentation predictions on the VOC cat dataset after finetuning on 131 training samples. The classification and segmentation model was pretrained on the dog images of the Oxford IIIT Pets dataset (Parkhi et al., 2012) whereas the self-supervised models were pretrained on the whole dataset.

that were pretrained on classification tasks to be less accurate on fine-grained details on the edge compared to models pretrained on segmentation/reconstruction tasks. This qualitative analysis underlines the reasoning that the classification-pretrained encoder is focussed on higher-level image characteristics.

## 5.2 Encoder Output Representations

With the above-mentioned findings, it is important to further study the representations that emerge when pretraining with a classification task, a segmentation task and self-supervised tasks. Therefore, we compare the representations of the models at the encoder output with each other. To compute the similarity between representations, we use *Center Kernel Alignment*. As we start training neural network models from scratch and the layer activations might be of a large dimension, it is a challenging problem to compare the representations of two models. Centered Kernel Alignment (CKA) is a method that can reliably identify correspondences between representations in networks trained from different initializations (Kornblith et al., 2019). CKA yields values between 0 and 1 where a larger value indicates more similarity between the representations

We consider the Pascal VOC 2012 dataset (Everingham et al., 2010) as test benchmark because it contains a variety of objects and scenes. We pretrain three different segmentation models on the benchmark, each having a different segmentation target and one classification model. As segmentation targets, we choose motorbike, person, and car and only show the images where the different classes are presented to the model. We provide example images, ground truth annotations and respective model predictions in Figure 7a.

For the classification model, we build a dataset with four different classes and the model has to distinguish between motorbikes, cars, cats, and airplanes. As this dataset contains the same images as the motorbike and car segmentation models, the models might develop similar representations during training.

After pretraining these models, we compute latent feature representations for all images from Pascal VOC, using the encoder output. We flatten the representation of size $8 \times 8 \times 512$ and then compare them pairwise between the different models with CKA. This yields a similarity matrix which is shown in Fig. 7. Larger values indicate that the representations are more similar. The results show that the different segmentation models (Car Segmentation, Person Segmentation, Motorbike Segmentation) have representations that are more similar to each other compared to the representations from the classification model. This indicates that although the decoder may have a negligible effect for downstream segmentation tasks, the encoder learns

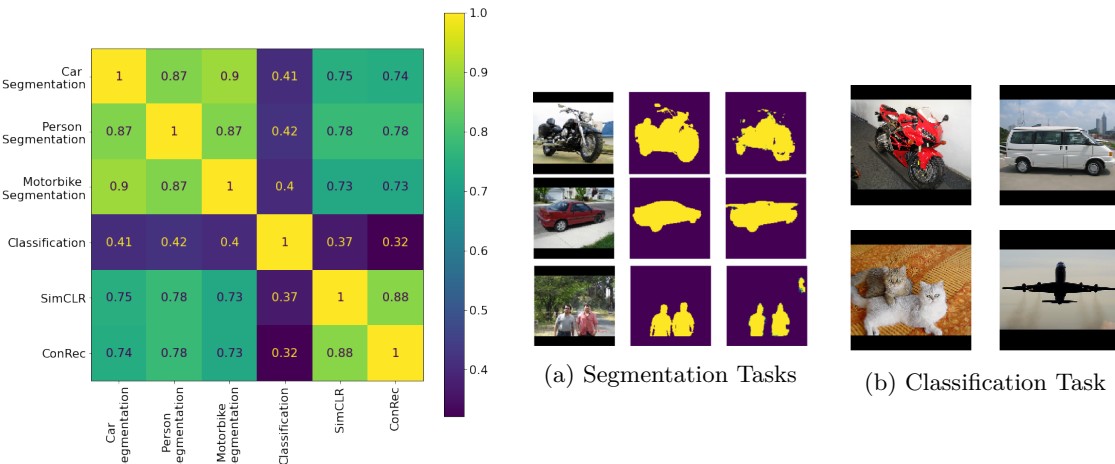

(a) Segmentation Tasks     (b) Classification Task

Figure 7: Representation similarity matrix between segmentation, classification and self-supervised models on the Pascal VOC dataset. We compare the representations with Center Kernel Alignment. Larger values indicate more similarity. (a) shows examples of the three segmentation tasks with respective predictions by the learned model. (b) displays one sample for each class of the 4-class classification problem.

fundamentally different representations than during a classification pretraining task. One simple explanation for this observation is that more spatial information is preserved at the encoder output, which is relevant for the segmentation learning task, but does not have any benefit for making classification decisions.

The self-supervised models SimCLR and ConRec have been pretrained on the training and validation split of the Pascal VOC 2012 dataset (Everingham et al., 2010) for 150.000 training steps. We can see that the self-supervised models all learn similar representations in comparison to the embeddings of the other models.

We were hoping to get more insights if the improved performance of ConRec is also highlighted in the representations as building more similar representations to a downstream task. However, this analysis does not show that the self-supervised representations are closer to the representations of a downstream classification or segmentation task. The similarity to the classification and segmentation models is not significantly different between the three self-supervised models.

## 6   Discussion

We examined the benefit of using pretrained encoders and decoders for image segmentation tasks. Our work is particularly focused on a binary segmentation scenario in the presence of a small number of labeled images. Given a U-Net architecture, we found out that there is little advantage in transferring weights from pretrained decoders: the convergence time and therefore ecological footprint gets reduced, however, the final downstream segmentation accuracy is not improved – even in low-data regimes. Only the encoder holds valuable weights that improve the final performance. Secondly, we compared the benefit of different pretext tasks, and we propose a contrastive self-supervised approach to pretrain encoders in the absence of labels (e.g. ConRec). We found that encoders trained on similar segmentation or reconstruction tasks perform better for downstream segmentation tasks than those trained on classification tasks.

We can conclude that encoders trained on different segmentation tasks are more similar to each other than to classification tasks, which we also demonstrated with a representation similarity analysis. Further, the effect of pretraining gets less significant when increasing the size of the finetuning dataset. It will be the subject of future work to validate these findings on different architectures besides U-Net and to include other self-supervised pretraining methods such as BYOL (Grill et al., 2020) and MoCo v3 (Chen et al., 2021) into the analysis. Further, we showed that adding multiple reconstruction tasks to the pretraining can improve downstream performance in segmentation tasks. Future work should study the contribution of each task in more detail. Also, our findings indicate that further experiments on large scale segmentation datasets can

provide encoder weights that are more suitable for downstream segmentation tasks than the weights from the ImageNet classification task.

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

## A ConRec Objective Ablation

To study the effect of the additional decoding tasks in the ConRec pretraining, we run an ablation experiment on the Oxford Flowers and Oxford Pets dataset with the same setup as in Table 2. Table 6 compares the vanilla ConRec implementation (Dippel et al., 2021) to our decoding task addition. The results show that in low data regimes, the additional decoding tasks provide a benefit in downstream task performance. We suppose that in order to solve multiple decoding tasks, the model has to learn more general representations that lead to better transferability of the model.

Table 6: ConRec decoding tasks ablation study. Finetuning results on the Oxford Flowers 17 and Oxford Pets datasets. Displayed values are the dice coefficient on the evaluation set.

| Reconstruction Tasks | Flowers | | | Pets | | |
|---|---|---|---|---|---|---|
| | 1% (8) | 5% (42) | 20% (170) | 1% (74) | 5% (368) | 20% (1475) |
| Reconstruction Task (Vanilla ConRec) | 82.5 ± 0.7 | 91.3 ± 0.2 | 94.8 ± 0.1 | 82.4 ± 0.4 | 87.1 ± 0.18 | 89.6 ± 0.1 |
| 4 Decoding Tasks (shown in Figure 3) | 85.7 ± 1.8 | 92.2 ± 0.4 | 95.3 ± 0.1 | 83.1 ± 0.5 | 87.5 ± 0.6 | 90.0 ± 0.1 |

