# OpenReview forum: "Transfer Learning for Segmentation Problems: Choose the Right Encoder and Skip the Decoder"
_TMLR — Rejected by TMLR_

### Review · Reviewer_dCau · 2022-08-02

**Summary Of Contributions:**

The paper studies the impact of pre-training for segmentation problems with encoder-decoder architectures. Specifically, the authors extend an existing self-supervised learning method ConRec with several pixel-wise pre-text tasks, e.g., predicting the normalized color, the original color, and segmentation of the masked patch. The paper argues that only the pre-trained encoder is useful when transferring to segmentation tasks while the pre-trained decoder doesn't bring improvements.

**Broader Impact Concerns:**

No.

**Requested Changes:**

- Experiments on more popular and challenging datasets.
- Experiments on more challenging segmentation tasks.
- Qualitative and quantitative analysis of the modified self-supervised learning method ConRec.
- Discussion and comparison with existing self-supervised learning methods for segmentation.


**Strengths And Weaknesses:**

## Strengths
- The motivation is clear. To study the impact of pre-trained encoder and decoder for downstream tasks like segmentation problems makes a lot of sense.
- The paper is organized. It is easy to follow and understand.

## Weaknesses
- All the experiments are conducted on tiny datasets with only a few hundred images, as listed in Table 1. It is unclear whether the conclusion still holds when using larger and more popular datasets, e.g., COCO and ADE20k.
- The authors only consider the simple segmentation tasks, e.g., flower segmentation. However, the community is more interested in more challenging segmentation problems like semantic segmentation and instance segmentation.
- The modified self-supervised learning method ConRec doesn't make sense. It's unclear how each prediction target contributes to the final results. In addition, the prediction targets of 1) highlighting the regions in the image where a mask can be found, and 2) filling the masks with black color, are somewhat unreasonable to me.
- In recent years, there are many self-supervised learning methods[A, B, C] are proposed for segmentation tasks. They should be discussed and considered.
- For pre-training, we aim to make it universal and general. Thus, the conclusion is not convincing when the experiments are on tiny datasets for very simple tasks.

[A]  Contrastive learning of global and local features for medical image segmentation with limited annotations
[B] Unsupervised learning of dense visual representations
[C] Dense contrastive learning for self-supervised visual pre-training

---

> ### Author Response · Authors · 2022-09-02
> **Additions to the Rebuttal**
>
> * The reasoning behind the additional decoding tasks is to learn more general representations. By having a diverse set of output reconstructions, the model has to remain more abstract representations until the last block of the decoder. The ablation in Table 6 shows that this leads to better performance in the downstream task. As the majority of the parameters from the decoder are shared, the additional decoder tasks only have a small overhead compared to the overall computation.
>
> * We focused with our analysis on the more commonly used SimCLR framework and its interplay with a reconstruction task. Unfortunately, we did not have time to run experiments with the mentioned segmentation-specific methods during the Rebuttal phase. We will discuss and cite the mentioned self-supervised approaches that are tailored for segmentation in the Related Work section.

---

> ### Comment · Action_Editors · 2022-09-22
> **About: Final recommendation**
>
> Dear Reviewer dCau,
>
> Thanks for reviewing the paper. Could you please help with adding the final comments, and final recommendation to the manuscript.  I suppose there is a recommendation button for accept/reject. Thanks!
>
> best,
>
> AE

---

> ### Comment · Reviewer_dCau · 2022-09-25
> **Response**
>
> Thanks for the rebuttal! After reading the rebuttal and the revised paper, my concerns still exist.
> - It's still unclear whether the conclusion still holds for larger and more challenging datasets like ADE20k and COCO, and segmentation tasks like semantic segmentation and instance segmentation.
> - Although the authors added Table 6 in the appendix in the revised version, it still cannot tell the contribution of each prediction target.

---

> > ### Author Response · Authors · 2022-09-26
> > **Response**
> >
> > Thanks for your comment!
> >
> > (1) We clearly state that our study is addressing the low-data scenario with a binary segmentation task (see Abstract and Discussion). We expect the benefit through pretrained models for other downstream segmentation tasks with more data to be low. Based on your feedback, we added an experiment with a pretraining on the SceneParse150 dataset (20k images), which is comparable to ADE20k (25k images).
> >
> > (2) the individual contribution of each training task is indeed an interesting question. We performed some initial experiments on this question and we found that for different downstream segmentation problems, different pretraining tasks were improving the performance. As there is no clear conclusion possible based on our datasets, we consider this analysis as an interesting research question for a follow-up investigation. To also highlight this in the paper, we added the following to the Discussion section:
> >
> > "Further, we showed that adding multiple reconstruction tasks to the pretraining can improve downstream performance in segmentation tasks. Future work should study the contribution of each task in more detail."

---

### Review · Reviewer_mQJJ · 2022-08-05

**Summary Of Contributions:**

This paper studies pretraining methods for the segmentation task, which generally uses the U-Net architecture with the pretrained encoder and randomly initialized decoder as the starting point of fine-tuning it on other downstream segmentation tasks. Specifically, the authors try to verify 1) which pretraining task is better among classification and segmentation in either supervised or self-supervised manners; 2) whether it would be good to pretraining both encoder and decoder (i.e., entire U-Net) or not. The authors further confirm the randomly initialized decoder in a segmentation-based pretrained U-Net can reach the fully-initialized model. Several methods, including SimCLR, reconstruction, and a variant of ConRec, are adopted for the study.

**Broader Impact Concerns:**

This paper does not seem to have any concerns about ethical or negative societal impact.

**Requested Changes:**

See first the above weaknesses and reflect them in the next revision.
- The authors specified "low-data regime" in many parts of the paper, but I recommend the authors involve more results pretrained on larger datasets. Because the datasets used in this paper are not actually used for pretraining generally. Shouldn't the low-data regime be applied only to fine-tuning? Please involve more downstream fine-tuning experiments with full datasets (not subsets):1) ImageNet-pretrained encoder; 2) Segmentation pretraining on the entire Cityscape or COCO2017 datasets.
- Please clarify what dataset is used for pertaining in Table 2.
- Please clarify the training setups for all the results for the different methods and scenarios.
- Can the authors provide the results of the reconstruction method in Figure 4. (a)?

**Strengths And Weaknesses:**

### Strengths
+ This paper is well written and quite easy to follow
+ The logical progression in this paper looks sound
+ Multiple runs of experiments make the experiments more convincing.

### Weaknesses
- The authors claimed that using ImageNet-pretrained models for segmentation tasks is suboptimal, unveiling it (in the introduction and discussion), but the authors did not actually experiment with ImageNet-pretrained models.
- All the experiments are performed with tailored datasets such as Cityscape Bus, which is a subset of the Cityscape dataset. I am not convincing that the authors intentionally performed experiments on low-data regimes. There is no way to determine if the authors experimented with small subsets due to some reasons and packaged them as low-data regimes or if that's really what he intended.
- No explanations why the subsets from the original dataets are chosen (e.g., in Table 1).
- U-Net has many variants and many encoder/decoder-based architectures, so I am not convincing that the experimental results can be generalized to any architecture.
- Randomly initializing the pretrained-decoder only can yield a similar result with fine-tuning with the fully initialized mode is surprising, but what's the advantage of this? Isn't reinitializing the decoder randomly after pertaining cumbersome?
- Training setups for the experiments are not specified. Additionally, I cannot identify whether the authors found the best training setups for each regime in tables or not. Please specify the training setups for all the experiments for different scenarios and different pertaining methods in each table.
- I cannot agree with the authors on why ConRec should be upgraded to be a more robust method in this analytical paper. Additionally, the competing SimCLR used in this paper is naively used without any modifications.
- Only SimCLR is adopted as a representative self-supervised method with class labels. Please involve further progressed methods such as BYOL, MoCOv3, or Ballow Twin.

---

> ### Author Response · Authors · 2022-09-02
> **Additions to the Rebuttal**
>
> * Subsets of the original datasets are chosen for finetuning to amplify the effect of the pretrained models. When using the complete dataset, the impact of the pretraining is small and the models perform almost equally, independent of the pretraining. We also added this to the description.
>
> * When pretraining the model on a segmentation/reconstruction task, valuable information for a downstream segmentation task can either be in the pretrained encoder or both blocks of the pretrained encoder-decoder.  By randomizing the decoder, we found out, that the pretrained encoder is the main driver. Thus, when pretraining with a segmentation/reconstruction task or with ConRec, we obtain a "better" encoder for the downstream task. Using a pretrained decoder is not too beneficial, as our experiments show that scenarios 3 and 4 obtain comparable results even in a low data regime.
>
> * During pretraining, SimCLR cannot be upgraded for a segmentation or reconstruction task, as SimCLR only trains an encoder with a contrastive loss. ConRec extends SimCLR with a decoder and a reconstruction loss next to the contrastive loss. Therefore, we can only upgrade ConRec with additional tasks on the decoder. Please also refer to the Table 6 in the Appendix that highlights the effect of the "upgrade" to 4 tasks compared to a single reconstruction task in the original ConRec implementation.
>
> * Indeed, we only studied SimCLR and no other self-supervised model. Studying BYOL and MoCov3 will be subject to future research. However, as they also train the encoder only, we assume that they might behave equivalent to SimCLR for downstream segmentation tasks. We also added this as part of our Discussion.

---

> > ### Comment · Reviewer_mQJJ · 2022-09-25
> > **Response to the rebuttal**
> >
> > Thanks for the response with the additional experiments and clarifications. I carefully read the authors' responses and the other reviewers' comments and feel some of my concerns were addressed. However, as other reviewers have pointed out (this reviewer also commented), this paper does not contain fine-tuning results trained on a complete dataset. The authors newly added the fine-tuning results using the ImageNet-pretrained backbones in Tables 4 and 5, but the experiments were still performed on sub-datasets; therefore, it is problematic for the authors' claim holds generally. Furthermore, I recommended the authors should perform more experiments with further other architectures over U-Net with more SSL methods because the argument is too broad without sufficient backups. I think the authors need to narrow down the tone of the claim only with the provided experiments.

---

> > > ### Author Response · Authors · 2022-09-27
> > > **Response**
> > >
> > > Thanks again for your feedback. Indeed, we already narrowed down the tone of our statement to not be broader than the experiments. In our recently uploaded version, we changed the tone of the abstract and the discussion. Based on your comment, we again changed the tone of our claims:
> > >
> > > In the Abstract:
> > >
> > > "We demonstrate that pretrained weights for a decoder may yield faster convergence..."
> > > -->
> > > "We exemplify within our experimentation framework that pretrained weights for a decoder may yield faster convergence..."
> > >
> > > "We find that transfer learning the decoder does not help downstream segmentation tasks, while transfer learning the encoder is truly beneficial."
> > > -->
> > > "Given a U-Net architecture, we find that transfer learning the decoder does not help downstream segmentation tasks, while transfer learning the encoder is truly beneficial."
> > >
> > >
> > > In the Discussion:
> > >
> > > “First, we found out that there is little advantage in transferring weights from pretrained decoders…”
> > > ->
> > > "Given a U-Net architecture, we found out that there is little advantage in transferring weights from pretrained decoders…”
> > >
> > > Further, we also state in the Discussion that it is important future work to validate our findings on other architectures and self-supervised learning tasks.

---

### Review · Reviewer_Bi3G · 2022-08-20

**Summary Of Contributions:**

This paper systemically evaluates and analyzes the effect of different pre-training and initialization approaches for encoder-decoder architectures on downstream segmentation tasks. It demonstrates that pre-training decoder may bring faster convergence speed, but not bring  improvement of segmentation performance compared to random initialization. Besides, it finds that pre-training the encoder on segmentation task is better than pre-training the encoder on the classification task, which is consistent with intuition to an extent. The authors also extend the ConRec approach with multiple reconstruction tasks and apply such method to the pre-training of segmentation network and evaluate its effectiveness.

**Broader Impact Concerns:**

I think this paper won't raise any serious ethical concerns.

**Requested Changes:**

I think the authors should make changes to address the concerns in the above weakness part.

**Strengths And Weaknesses:**

Pros:
- Generally speaking, the paper is easy to follow.
- The findings are interesting and to some extent consistent with the intuition.

Cons:
- Some key details are missing and not clearly described. In Table 2\&4, actually I don't know what each scenario exactly represents. I didn't find any clear descriptions about this.  Besides, what does the numbers represent in all the tables, e.g. mIou or pixel-wise accuracy? And I don't understand why the authors use dice coefficients to represent the performance of a segmentation model, rather than the common evaluation metric on validation/test dataset, e.g. mIoU.

- I think the current experiment results are not enough to support the authors' claim. The authors just perform evaluations on relatively easy segmentation tasks, i.e. foreground/background separation. How about segmenting multiple objects in one image? Besides, the authors evaluate the model on relative small datasets, e.g. at most 20\% of original dataset. Will the conclusions hold if we own large amounts of annotations to fine-tune our segmentation model?

- I don't think the authors' results challenge the important role of pre-training encoder on ImageNet dataset (or large-scale classification dataset). The authors do compare the pre-training on classification task and pre-training on segmentation task, and demonstrate pre-training on segmentation task is better. But the amount of data used for pertaining with classification task is quite small and the classification task is quite easy, e.g. the number of classes is quite limited (e.g. 1). In my opinion, we pre-train the encoder with classification task in order to encode better semantics. When the amount of data or the complexity of classification task is limited, it is natural that we cannot reach our goal. For pre-training with segmentation task, I think it may work because it enable features to encode more rich spatial details. Both the semantic information and the spatial details are important for segmentation. So I don't agree with the authors that pre-training on segmentation task may be generally better than pre-training on classification task.

- The authors claims that they extent original ConRec approach to make it more suitable for segmentation pre-training. I think the justification for this is not enough.

---

> ### Author Response · Authors · 2022-09-02
> **Additions to the Rebuttal**
>
>
> * We agree that in the previous version of the manuscript, it was not easy to understand which exact scenario was assessed in each Table. Therefore, we added more context to the corresponding Tables.
>
> * Indeed, our analysis is targeting binary segmentation scenarios with small datasets for training/finetuning. We do not claim that our findings hold in the presence of large datasets, as overall the positive effect of pretraining gets less significant when increasing the size of the finetuning dataset. We also state this in the Discussion.

---

### Review · Reviewer_ru6t · 2022-08-27

**Summary Of Contributions:**

The authors perform an exploration of self-supervised learning for segmentation tasks. In segmentation tasks, it is common to have an "encoder" and also "decoder" parts of the network, and interestingly, the authors show that it does not help to pretrain the decoder part of the network as well.
The authors consider both supervised training, and also self-supervised pre-training based on the ConRec method of Dippel et al (and also the authors extension of this method).

**Broader Impact Concerns:**

I do not see any issues.

**Requested Changes:**

As mentioned in the "Weaknesses" above, the paper would be signficantly stronger with experiments on standard experimental set-ups used in the Computer Vision community. As this is largely an empirical study, it is not clear to me that the findings would still hold on larger datasets and the standard experimental set-ups used by the community.

Moreover, I would suggest to include additional ablation studies showing that each of the additoinal tasks added by the authors to the ConRec algorithm does indeed performance and is necessary.



**Strengths And Weaknesses:**

### Strengths

The authors perform fairly thorough analyses of pretraining methods for segmentation networks, in a number of settings (ie pretraining the encoder, pretraining the encoder and decoder).

The authors also show that different pretraining methods produce different representations in the encoder of the network with Center Kernel Alignment.

It is also useful practically to see that pretraining the decoder with self-supervised methods does not provide any benefit over simply randomly initializing the decoder for downstream segmentation tasks.

The authors extension of the ConRec method also performs better than the original method.


### Weaknesses


Although the authors have performed numerous experiments, a weakness of the paper is that none of the experiments are on standard segmentation datasets. The authors have only used very small datasets like Oxford Flowers and Pets. And when they have used more standard datasets like Pascal VOC and Cityscapes, they have only used very subsampled versions of the dataset. Therefore, it is not clear to me that the findings of the authors will be observed on the more real-world datasets used by the computer vision community.

The authors also claim that "the common practice of using ImageNet-pretrained weights for segmentation tasks is suboptimal". Whilst the authors experiments suggest this, this could be validated by actually doing experiments with ImageNet-pretrained models; and comparing standard supervised pretraining on ImageNet to self-supervised learning instead.

The arguments made by the authors would be significantly stronger if the experiments were on larger, real-world datasets, and the standard experimental set-up used by the community.

The authors also propose an extensions of the ConRec method of Dippel et al by adding multiple pretext tasks. However, the authors do not actually show if each of these additional pretext tasks is really necessary or not. Some additional ablations here would be welcome, and make the paper stronger.



Minor typos:

- Abstract: "the finding implicates that ..." -> "implies that ..."
- Section 2.2: "fain-grained" -> "fine-grained"
- Section 2.2. "The four tasks include ..." and then only three tasks are described.

---

### Author Response · Authors · 2022-09-02
**Rebuttal**

Firstly, we would like to thank the four reviewers for the time and effort to review our manuscript and for the constructive feedback which was provided. Overall, the comments shared four common points that motivated us to run additional experiments and adjust the presentation of the results.

**Imagenet Comparison**

Reviewers ru6t, Bi3G, and mQJJ asked to include a baseline with an ImageNet initialization. We addressed this point by including an ImageNet baseline in Table 4 and Table 5. We use a Resnet18 encoder with a random decoder from the segmentation_models library (https://github.com/qubvel/segmentation_models).

**Pretraining Dataset Size**

The reviewers raised the concern that the pretraining dataset might be too small. Therefore, we ran additional experiments and pretrained a U-Net model on the MIT Scene Parsing Benchmark (SceneParse150) [1] which contains the most common 150 classes of ADE20k as labels and 20,210 training images. Finetuning results of this model are provided in Table 4 and Table 5 in the manuscript. Further, we want to highlight that while the finetuning datasets are small, the Oxford Pets dog pretraining dataset also contains 5k images which is a significant size for segmentation. Also, note that we have been describing a real-world segmentation scenario, recently published in the medical imaging domain in Section 4 (Implication for Real World Scenario). Especially segmentation problems in the medical domain suffer from small datasets that are available for finetuning, which exemplifies the relevance of our work.

**ConRec Ablations**

Reviewers ru6t, Bi3G, mQJJ raised the point that there is no ablation given for ConRec's loss design. We address this point by comparing the presented ConRec variant with multiple reconstruction heads to the original ConRec implementation. Table 6 in the Appendix demonstrates the superiority of multiple reconstruction tasks over the original implementation.

**Dataset Selection and Low data regime**

Several Reviewers (mQJJ, dCau) have raised the concern that the finetuning experiments were not conducted on large-scale benchmark segmentation datasets. Indeed, our experiments were motivated by medical image segmentation, where labeled data is scarce, and the binary segmentation case is more common. Thus, the binary segmentation problem with only a few labeled images was replicated by using subsets (1\%, 5\%, 20\%) of well-studied segmentation benchmark data (i.e. Flowers, Pets etc.)
We further highlighted in the manuscript that our work is particularly focused on a binary segmentation scenario in the presence of a small number of labeled images (see Discussion).

[1] Zhou, Bolei, et al. "Scene parsing through ade20k dataset." Proceedings of the IEEE conference on computer vision and pattern recognition. 2017.

---

### Decision · Action_Editors · 2022-10-04

**Recommendation:** Reject

**Comment:**

There are several points that are pointed by reviewers and should be significantly addressed:

1,  it is advisable to either narrow down the tone of the claim only with the provided experiments, or do more experiments to avoid the overstatement. Particularly,

(1.1) this paper does not contain fine-tuning results trained on a complete dataset. The paper contains the newly added the fine-tuning results using the ImageNet-pretrained backbones in Tables 4 and 5, but the reviewers suggested the experiments were still performed on sub-datasets; therefore, it is problematic for the claim holds generally.

(1.2) it is advisable to perform more experiments with different other architectures over U-Net with more SSL methods, as the argument is too broad without sufficient backups.

(1.3) the reviewers are still unclear whether the conclusion still holds for larger and more challenging datasets like ADE20k and COCO, and segmentation tasks like semantic segmentation and instance segmentation.

(1.4) On the other hand, the reviewers admit the merit of the submission that the method works well on small-scale experiments (more akin to the medical imaging scenario) which reduces its applicability to the larger computer vision community.

2, About novelty comments from the reviewers:
As the extension of ConRec is quite minor,  it may be advisable to further improve the novelty of this paper (if possible).

3, Minors:
The reviewers still have some minor comments which shall be addressed:

(3.1) why the authors use dice coefficients to represent the performance of a segmentation model, rather than directly evaluating mIoU on validation/test dataset.

(3.2) Although Table 6 is added in the appendix in the revised version, it still cannot tell the contribution of each prediction target. It should be further clarified in the submission.

As the TMLR does not have the choice of major revision, the AE suggested that the manuscript should go through a more thorough revision (bigger than minor revision) before it can be accepted to TMLR.

**Audience:**

This paper has good potential of being benefit to many TMLR's audience; the methods and findings may benefit the researchers who are working on image segmentation.

**Claims And Evidence:**

This paper is reviewed by four experts on the related topics. The review process takes a bit longer than expected;
There are several rounds comments and rebuttals between authours and reviewers.
As for the final recommendation, three reviewers vote the rejection, and one reviewer vote the acceptance. However, all the reviewers actually almost concern the same problem existed in this paper:
Though ''the paper is technically sound for the most part; and the experimental analyses would have benefits to some readers'',
''the augments and claims of this submission is too broad without sufficient backups''. It is thus suggested to
''narrow down the tone of the claim only with the provided experiments or do more experiments to avoid the overstatement.''.
Please refer to the TMLR's evaluation criteria for more details on this point.